# Ultra-Broadband Ultraviolet–Visible Light–Short Wavelength Infrared InGaAs Focal Plane Arrays via n-InP Contact Layer Removal

**DOI:** 10.3390/s24051521

**Published:** 2024-02-26

**Authors:** Jiaxin Zhang, Wei Wang, Haifeng Ye, Runyu Huang, Chen Liu, Weilin Zhao, Yanli Shi

**Affiliations:** 1School of Physics and Astronomy, Yunnan University, Kunming 650504, China; zhangjiaxin@ghopto.com (J.Z.); wangwei@mail.ynu.edu.cn (W.W.); yhf123@mail.ynu.edu.cn (H.Y.); huangry1998@163.com (R.H.); 799936211@mail.ynu.edu.cn (C.L.); zhaoweilin@mail.ynu.edu.cn (W.Z.); 2Shanxi Guohui Optoelectronic Technology Co., Ltd., Taiyuan 030006, China

**Keywords:** InGaAs FPAs, n-InP contact layer, response spectrum, UV–VIS–SWIR, quantum efficiency

## Abstract

PIN InGaAs short wavelength infrared (SWIR) focal plane array (FPA) detectors have attracted extensive attention due to their high detectivity, high quantum efficiency, room temperature operation, low dark current, and good radiation resistance. Furthermore, InGaAs FPA detectors have wide applications in many fields, such as aviation safety, biomedicine, camouflage recognition, and infrared night vision. Recently, extensive research has been conducted on the extension of the response spectrum from short wavelength infrared (SWIR) to visible light (VIS) through InP substrate removal and reserving the n-InP contact layer. However, there is little research on the absorption of InGaAs detectors in the ultraviolet (UV) band. In this paper, we present an ultra-broadband UV–VIS–SWIR 640 × 512 15 μm InGaAs FPA detector by removing the n-InP contact layer in the active area and reserving the InP contact layer around the pixels for n contact, creating incident light to be directly absorbed by the In_0.53_Ga_0.47_As absorption layer. In addition, the optical absorption characteristics of InGaAs infrared detectors with and without an n-InP contact layer are studied theoretically. The test results show that the spectral response is extended to the range of 200–1700 nm. The quantum efficiency is higher than 45% over a broad wavelength range of 300–1650 nm. The operability is up to 99.98%, and the responsivity non-uniformity is 3.28%. The imaging capability of InGaAs FPAs without the n-InP contact layer has also been demonstrated, which proves the feasibility of simultaneous detection for these three bands.

## 1. Introduction

P-type doping-intrinsic-n-type-doping (PIN) In_0.53_Ga_0.47_As/InP back-illuminated focal plane arrays (FPAs) have attracted extensive attention because of their low dark currents, high detectivity, high quantum efficiency, and good anti-radiation characteristics in uncooled conditions [1,2]. In addition, InGaAs FPA detectors have been widely used in applications in aviation safety, biomedicine, camouflage recognition, night vision, and other fields [3,4,5]. Recently, extensive research has been conducted on the extension of the response spectrum from short wavelength infrared (SWIR) to visible light (VIS) with InP substrate removal [6,7,8,9,10,11,12,13,14,15,16,17,18,19,20,21]. Because of the high absorption coefficient of n-InP in the visible light band and the different thicknesses of the n-InP contact layer [22], there are different spectral responses of VIS–SWIR InGaAs FPAs, such as 0.7–1.7 μm [6], 0.5–1.7 μm [7,8,9,10,11,12,13,14,15,16,17], and 0.4–1.7 μm [18,19,20,21]. Teledyne Judson Technologies fabricated VIS–SWIR InGaAs with a very thin n-InP contact layer, and the minimum quantum efficiency (QE) of above 40% was achieved in the entire visible wavelength range [14]. He Wei et al. presented a broadband high QE VIS–SWIR InGaAs detector by thinning an n-InP contact layer to 10 nm, and the quantum efficiencies were higher than 60% over a broad wavelength range of 500–1700 nm [15]. Sony Semiconductor Solutions Corporation used the complementary metal oxide semiconductor image sensor (CIS) wafer thinning technique with precise control over the thickness of an n-InP contact layer, resulting in an extremely thin n-InP contact layer and a higher QE than those of a conventional sensor in the 400–1700 range [21]. However, all of the above studies mainly focused on extending and improving the response spectrum of an InGaAs FPA detector in the visible light band by thinning the n-InP contact layer; few research studies have been conducted on the effect of the InP contact layer on the absorption of InGaAs infrared detectors in the ultraviolet band.

The utility of detecting light from ultraviolet (UV) to SWIR stems from the range of chemical bond energies and the corresponding photoluminescence and absorption spectra of typical materials. For example, organic compounds with conjugated carbon bonds display strong absorption at wavelengths ranging from 0.2 to 0.8 μm, whereas small carbon-based molecules, such as CH_4_, CO, and CO_2_, have spectral signatures in the SWIR range. The determination of various air pollutants and greenhouse gases through remote optical spectroscopy requires UV-SWIR imaging. Advances in UV-SWIR imaging also impact the detection of X-rays, gamma rays, and high-energy particles by expanding the range of usable scintillators, thereby contributing to biomedical applications, astronomy, and high-energy physics. In addition, separate image sensors are used for different spectral sub-bands: GaN for UV, Si for visible, and InGaAs for IR, requiring expensive component-level integration for hyper-spectral imaging [23]. These complications can be circumvented by a single image sensor that can cover the entire UV-to-SWIR spectrum.

Therefore, there is a need to further extend and enhance the response spectrum of VIS–SWIR InGaAs detectors to the UV band. In this paper, a novel ultra-broadband UV–VIS–SWIR 640 × 512 15 μm InGaAs FPA is presented. The n-InP contact layer in the active area is removed with the wet etching process, and the n-InP contact layer around the pixels is only reserved for n contact. The incident light is directly absorbed by the In_0.53_Ga_0.47_As absorption layer. The experimental results show that the spectral response is extended to 200–1700 nm, and the QE is higher than 45% over a broad wavelength range of 300–1650 nm.

## 2. Materials and Methods

### 2.1. Calculation of Optical Absorption

Reflection occurs when light propagates in the interface of the materials. For normal incidence, the reflectivity *R* is expressed as shown in Equation (1):(1)$$R=\frac{({n_{1}-n_{2})}^{2}+k^{2}}{\;\;\;({n_{1}+n_{2})}^{2}+k^{2}}$$

Light is absorbed when it propagates into semiconductor materials, and light absorptivity is expressed as shown in Equation (2):(2)$$P_{ab}=1-e^{-\alpha d}$$

Therefore, the light absorptivity is expressed as shown in Equation (3):(3)$$P_{ab}^{\prime }=(1-R)(1-e^{-\alpha d})$$
where *n*_1_ and *n*_2_ are the refractive indices of the two materials, *k* is the extinction coefficient of the light absorption material, *α* is the absorption coefficient, *d* is the material’s thickness, and *P_ab_* is the absorptivity.

The absorption coefficients, refractive index, and extinction coefficients of n-InP and In_0.53_Ga_0.47_As are calculated with linear fitting [22,24], as shown in Figure 1a–c, respectively. The absorption coefficient of InP in the ultraviolet band is higher than that in the visible range, as shown in Figure 1a. Furthermore, we calculated the absorptivity of InGaAs detectors without an InP contact layer and with a 10 nm InP contact layer; the results show that the mean absorptivity with a 10 nm InP contact layer is 15% in the UV band, while the mean absorptivity without an n-InP contact layer is up to 49% in the same band, increasing twice, as shown in Figure 1e. In addition, in the SWIR band, due to the refraction index of In_0.53_Ga_0.47_As being higher than that of n-InP, as shown in Figure 1b, the reflectivity of the n-InP contact layer is lower than that of In_0.53_Ga _0.47_As, as shown in Figure 1d, and the absorptivity of the detector with the InP contact layer is higher than that of the detector without the InP contact layer, as shown in Figure 1e.

### 2.2. Fabrication of InGaAs FPAs

The PIN InP/InGaAs epitaxial material comprises a 1 μm n-InP top layer, a 2.5 μm InGaAs absorption layer, a 0.2 μm n-InP contact layer with a doping concentration of 2 × 10^18^ cm^–3^, a 0.1 μm InGaAs stop-etching layer, and a 625 μm InP substrate layer. The SiN layer is deposited on the EPI wafer with PECVD. Then, the Zn diffusion hole is fabricated with SiN etching. Zn diffusion is carried out with Zn_3_P_2_ in a sealed tube. After that, the SiN layer is deposited on the wafer for Zn activation anneal, and the SiN is dry etched with reactive ion etching (RIE). The p-type metal is evaporated with an E-beam. Then, the InP and InGaAs around the active region are etched to expose the n-InP contact layer with wet etching, and the n-type metal is evaporated with an E-beam. The p-type metal and n-type metal are annealed for ohmic contact. The SiN layer is deposited for InP and InGaAs sidewall passivation, and the SiN on the p-type metal and n-type metal is dry etched with RIE. The connecting metal is deposited with an E-beam. The indium bump is deposited with a thermal evaporator. The InGaAs photodiode arrays (PDAs) are hybrid-integrated with a readout integrated circuit (ROIC) using the flip–chip bonding process after wafer dicing. After this, the underfill process is carried out to improve reliability. The substrate is removed with polishing and wet etching, and then, the InGaAs stop-etching layer is removed with wet etching, as shown in Figure 2a. SiN is deposited on the n-InP contact layer for passivation. The SiN in the active area is dry etched after photolithography, exposing the n-InP contact layer, and the n-InP contact layer in the active area is etched with wet etching, as shown in Figure 2b. SiN is deposited on the InGaAs absorption layer for passivation, as shown in Figure 3.

## 3. Results and Discussion

The InGaAs FPA detector is measured with a Pulse Instruments 7700 FPA test system and the EMVA 1288 [25] standard at room temperature. The light source is a halogen lamp, providing UV, visible light, and SWIR bands. We use grating to split the light source, and the bandwidth is 50 nm. An integrating sphere provides the uniform light signal received by the InGaAs detector. The QE, dark current density, operability, detectivity, and response non-uniformity are measured under medium gain.

### 3.1. Quantum Efficiency (QE)

A grating splitting method is used to measure the QE. We obtained QE_b_ from the FPA detector before n-InP contact layer etching and QE_a_ from the FPA detector after n-InP contact layer etching, as shown in Figure 4. Because the n-InP contact layer is removed, eliminating the absorption of UV and visible light by InP, QE_a_ is extended to 200–1700 nm, and QE_a_ is higher than QE_b_ in the UV and visible light bands. However, QE_b_ is higher than QE_a_ in the range of 1000–1700 nm, which is because In_0.53_Ga_0.47_As has a higher refractive index than that of n-InP, causing the higher reflectivity of In_0.53_Ga_o.47_As compared to that of InP/In_0.53_Ga_0.47_As. In addition, due to the increase in the extinction coefficient of In_0.53_Ga_0.47_As from 400 nm to 250 nm, leading to an increase in reflectivity, QE_a_ plunges from 400 nm to 250 nm; meanwhile, QE_a_ increases at 250–200 nm because the extinction coefficient falls. The QE_a_ at 200 nm, 300 nm, and 400 nm is 38.4%, 47.6%, and 57.7%, respectively. The maximum QE is 68.7% at 750 nm.

### 3.2. Responsivity Non-Uniformity

Before n-InP contact layer etching, the responsivity non-uniformity is 2.8%, as shown in Figure 5a. After n-InP contact layer etching, the responsivity non-uniformity is 3.28%, which is 17% higher than that with the n-InP contact layer, as shown in Figure 5b. The main reason for this is that the In_0.53_Ga_0.47_As absorption layer is stained with photoresist to protect the ROIC pad when the n-InP contact layer is etched in an acidic solution, as shown in the red box in Figure 5b.

### 3.3. Dark Current Density and Dark Noise

Before n-InP contact layer etching, the dark current density is 8.4 nA/cm^2^ at −0.1 V. After n-InP contact layer etching, the dark current density is 21.8 nA/cm^2^ at −0.1 V, which is 2.6 times as much as that with the n-InP contact layer, as shown in Figure 6. The increase in dark current is due to the surface leakage current from the poor interface between InGaAs and the upper passivation layer.

The grey value distributions of dark signal (noise) are shown in Figure 7, in which the horizontal axis is the grey value, and the smaller the grey value, the stronger the dark signal. After n-InP contact layer etching, dark signal increases from 2.47 mV to 3.24 mV; correspondingly, the noise electron increases from 98 e^-^ to 136 e^-^. The main reason for this is that the surface contamination of the InGaAs absorption layer during the process affects the passivation effect of silicon nitride on the absorption layer, and leakage channels are formed in the interface of the passivation layer and the absorption layer, resulting in an increase in surface leakage current.

### 3.4. Operability

The pixels with a responsivity lower than 30% of the average value are defined as dead pixels. The operability of the detector with the n-InP contact layer is 99.98%, and the number of dead pixels is 66, as shown in Figure 8a. After n-InP contact layer etching, the operability is almost the same as before etching, and the number of dead pixels is 72, increasing by 6 dead pixels compared to the detector with the n-InP contact layer, as shown in Figure 8a,b, in which the dead pixel distributions are also depicted in the inserted figures for comparison, with the white areas being dead pixels. The results indicate that the operability hardly deteriorates after n-InP contact layer etching.

### 3.5. Detectivity

We obtained the detectivity at 1550 nm, whereby the integration time is 6 ms and the light intensity is 0.5 μW/cm^2^. Before and after n-InP contact layer etching, the detectivity is 5.11 × 10^12^ cm·Hz^1/2^/W and 2.28 × 10^12^ cm·Hz^1/2^/W, respectively. There are two reasons for the decrease in detectivity after n-InP contact layer etching: one is the increase in dark noise, and the other is the decrease in quantum efficiency in the infrared band.

### 3.6. Imagery

Three light sources, including a 1550 nm LED, visible light, and a 254 nm LED are used as different imaging targets, while an InGaAs camera with an n-InP contact layer and an InGaAs camera without an n-InP contact layer are applied to image the three targets. The 1550 nm light and visible light are captured by the InGaAs camera with an n-InP contact layer, while the 254 nm light cannot be captured by the same InGaAs camera, as shown in Figure 9a. However, the strong light signals from three light sources are detected by the InGaAs camera without an n-InP contact layer, as shown in Figure 9b. In addition, we obtained a very clear image with the InGaAs camera without an n-InP contact layer under 254 nm in the lab, as shown in Figure 9c, demonstrating that the camera has strong imaging ability for the deep-UV band. It is indicated that the InGaAs detector works normally, and the spectral response is extended to the UV band with high performance after n-InP contact layer removal, which provides a foundation for expanding the application field of InGaAs detectors. 

## 4. Conclusions

In summary, we have proposed and verified a novel ultra-broadband UV–VIS–SWIR InGaAs FPA detector by removing the n-InP contact layer in the active area and reserving only the n-InP contact layer around the pixels for n contact. The light absorption characteristics of detectors with an n-InP contact layer and without an n-InP contact layer have been analyzed theoretically. The results reveal that the spectral response is extended to 200–1700 nm by removing the n-InP contact layer for the InP/InGaAs detectors. The QE is higher than 45% over a broad wavelength range of 300–1650 nm, and the QEs at 200 nm, 300 nm, and 400 nm are 38.4%, 47.6%, and 57.7%, respectively. In addition, the operability is up to 99.98% for 640 × 512 15 μm InGaAs FPAs, the responsivity non-uniformity is 3.28%, and the dark current density is 21.8 nA/cm^2^ at room temperature. These research results provide feasibility for the application of InGaAs detectors in the three bands of UV, visible light, and SWIR.

## Figures and Tables

**Figure 1 sensors-24-01521-f001:**
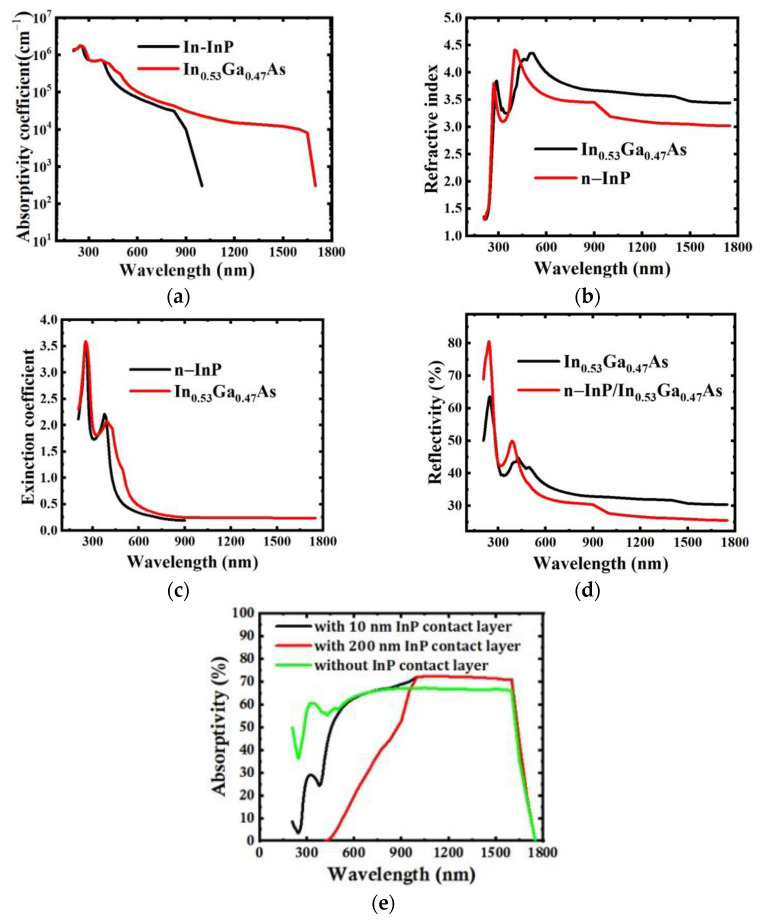
Parameters of n-InP and In_0.53_Ga_0.47_As: (**a**) absorption coefficients, (**b**) refractive index, (**c**) extinction coefficients, (**d**) simulated reflectivity of InGaAs and InP/InGaAs, and (**e**) simulated absorptivity of InGaAs detectors with three different fabrications.

**Figure 2 sensors-24-01521-f002:**
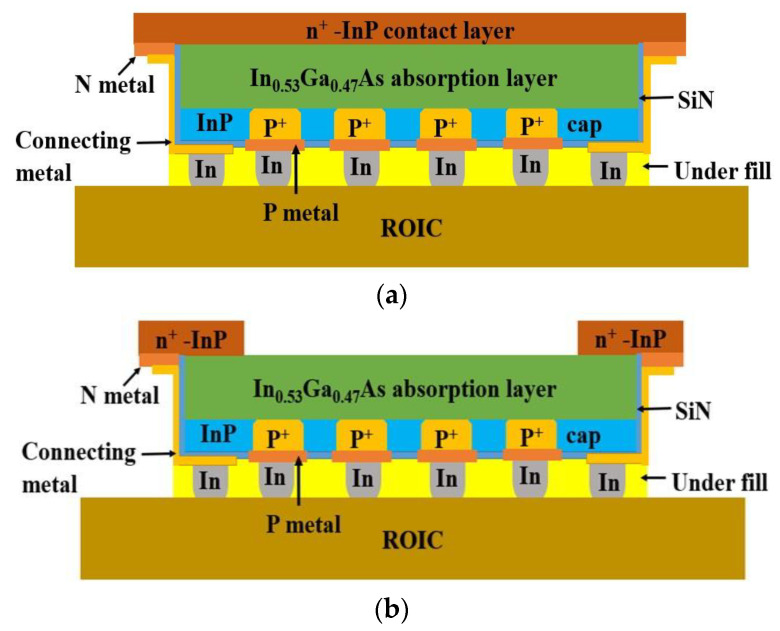
Schematic cross-section of the InGaAs FPA (**a**) before n-InP contact layer etching and (**b**) after n-InP contact layer etching.

**Figure 3 sensors-24-01521-f003:**
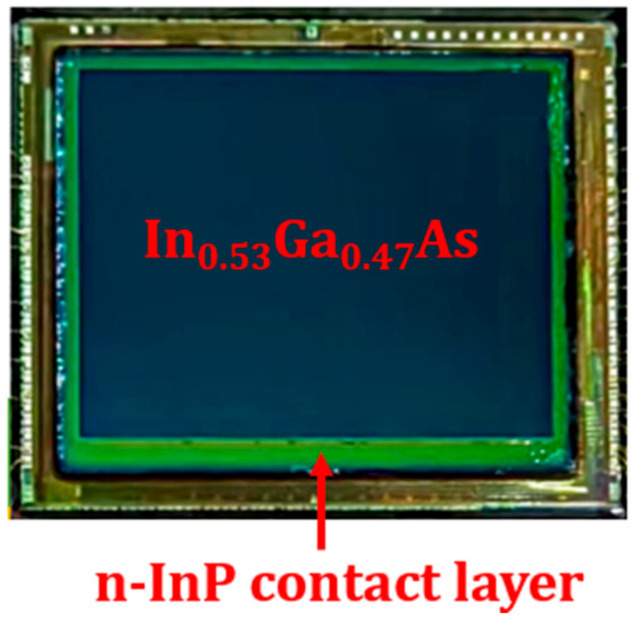
Top view of the InGaAs FPA after SiN passivation.

**Figure 4 sensors-24-01521-f004:**
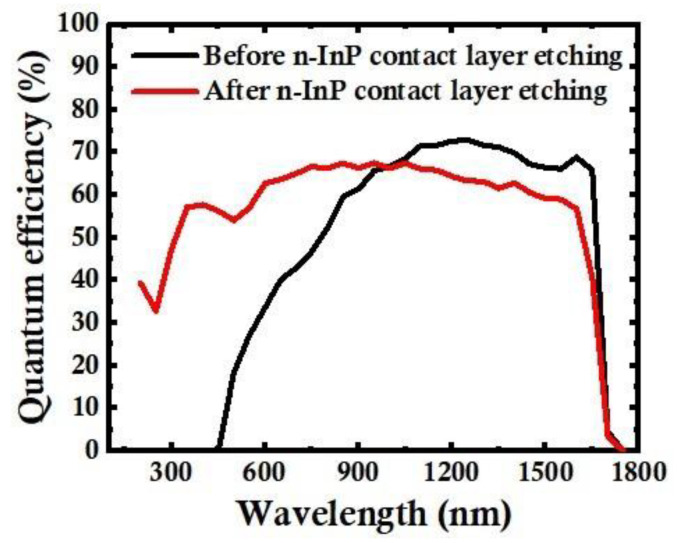
QE before and after 0.2 μm n-InP contact layer etching.

**Figure 5 sensors-24-01521-f005:**
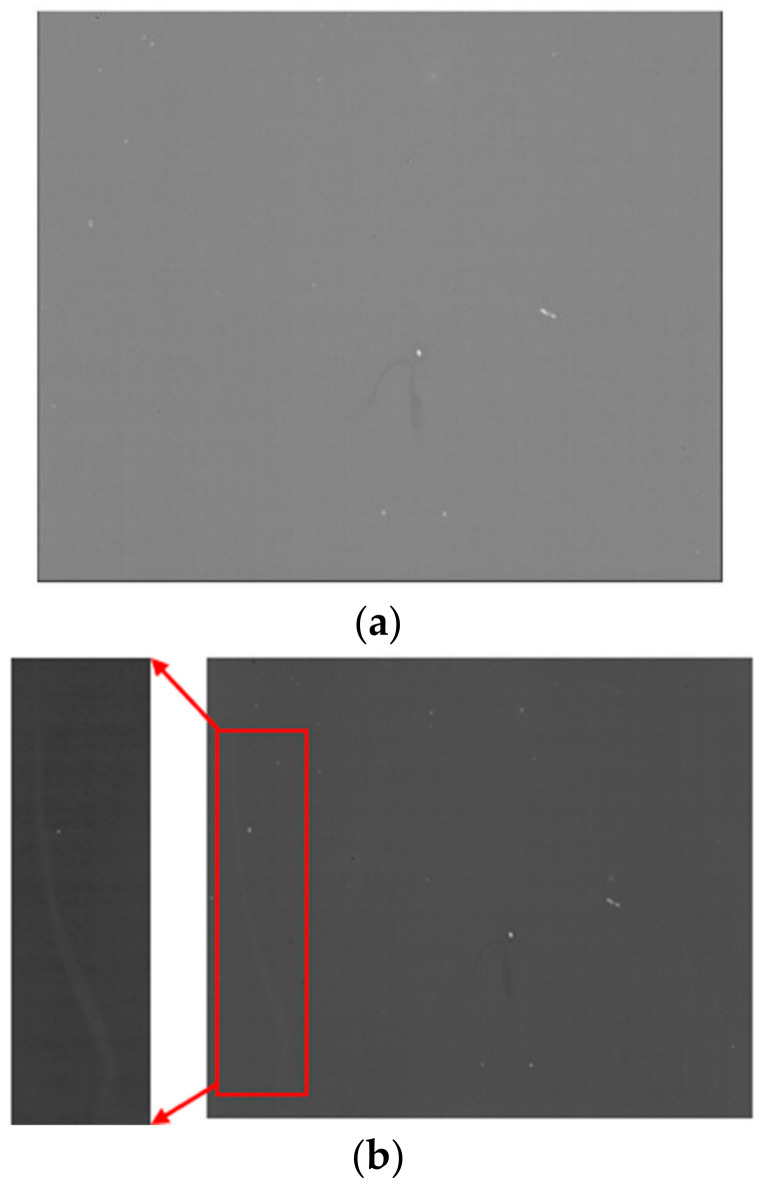
Response mapping in half saturation (**a**) before n-InP contact layer etching and (**b**) after n-InP contact etching.

**Figure 6 sensors-24-01521-f006:**
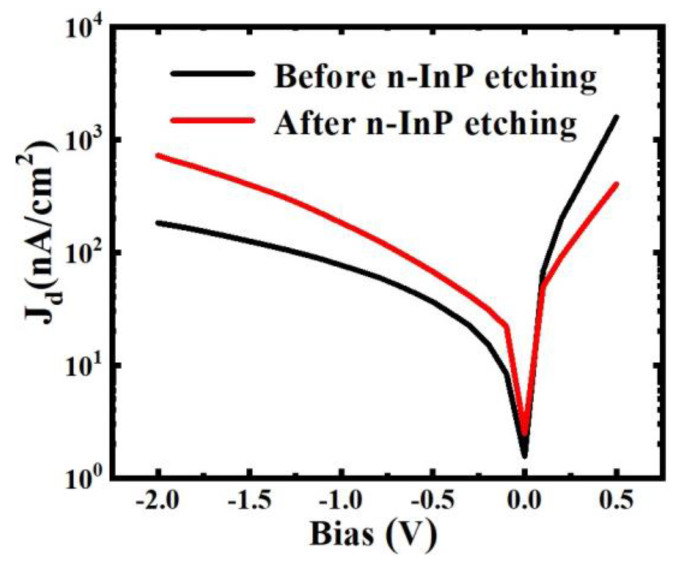
Dark current density before and after n-InP contact layer etching.

**Figure 7 sensors-24-01521-f007:**
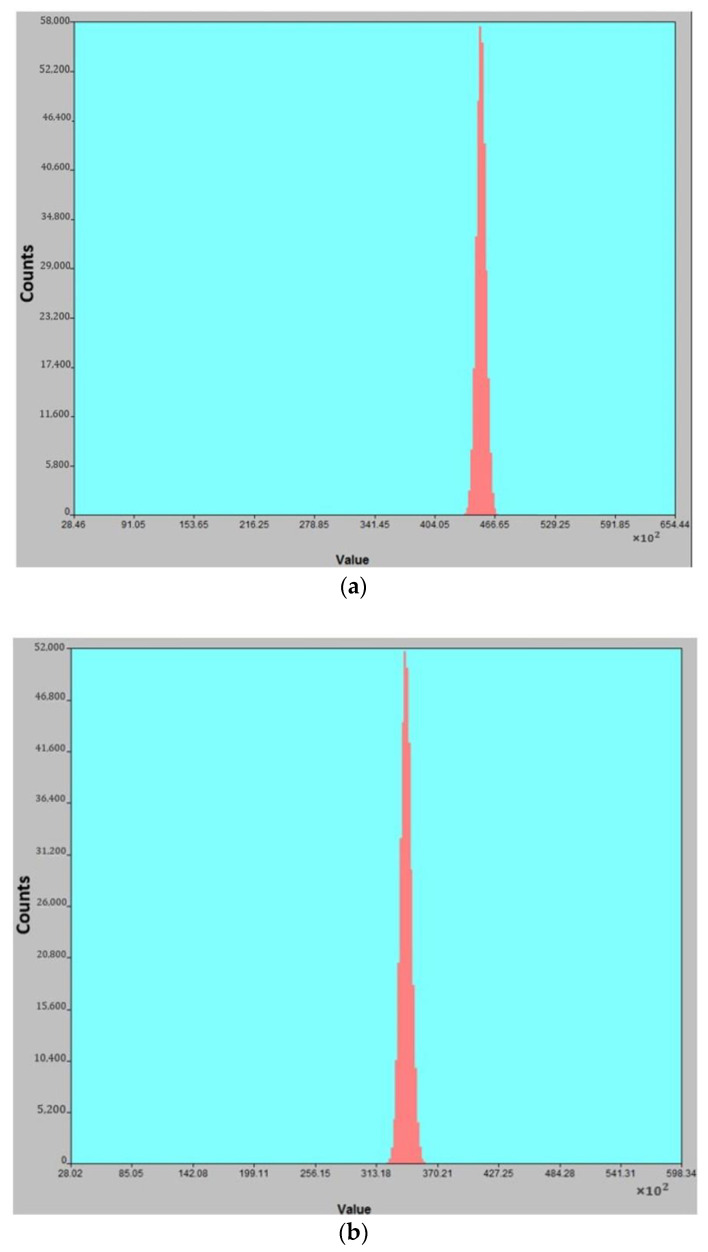
Distributions of dark signal (**a**) before n-InP contact layer etching and (**b**) after n-InP contact layer etching.

**Figure 8 sensors-24-01521-f008:**
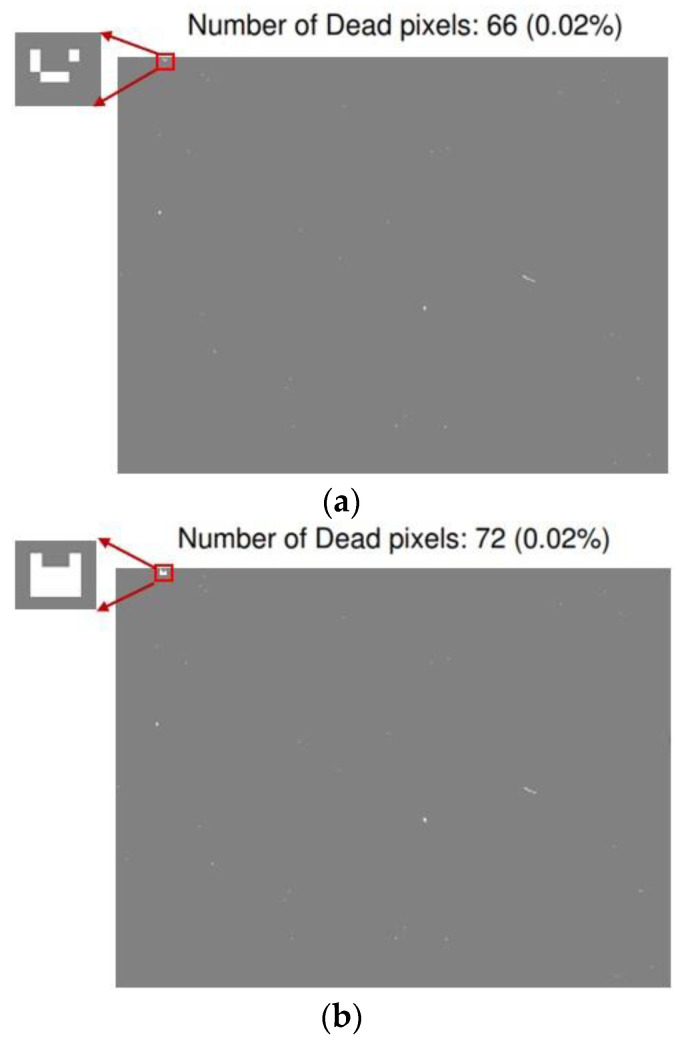
Operability mapping (**a**) before n-InP contact layer etching and (**b**) after n-InP contact layer etching.

**Figure 9 sensors-24-01521-f009:**
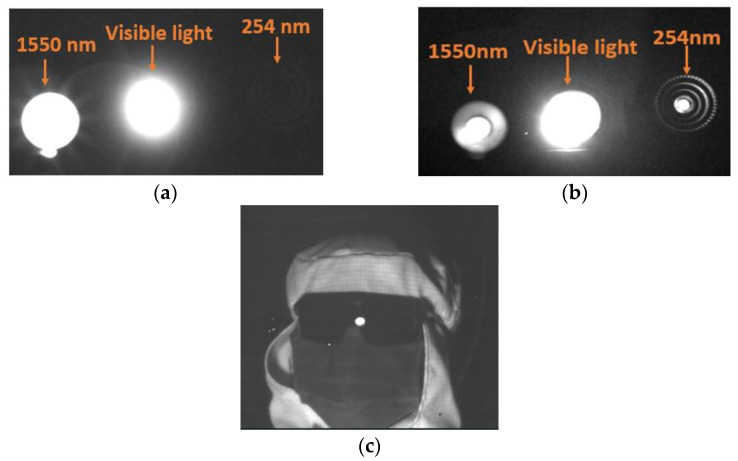
Images with different imagers: (**a**) InGaAs camera with an n-InP contact layer and (**b**,**c**) InGaAs camera without an n-InP contact layer.

## Data Availability

Data are contained within the article.

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
