# Peer review of "Ultra-Broadband Ultraviolet–Visible Light–Short Wavelength Infrared InGaAs Focal Plane Arrays via n-InP Contact Layer Removal"

_sensors, 2024, doi:10.3390/s24051521_

Round 1

Reviewer 1 Report

Comments and Suggestions for Authors

The authors have demonstrated a detector array based on InGaAs with a broad range sensitivity extending from 200 to 1700 nm.  This is accomplished by removing the InP substrate and passivating the surface of the active material.  They have used good processing methods and the testing is appropriate.

The manuscript will be good for publication after the following minor revisions.

1. The authors should examine all of the reference numbers in the text to see if they are correct, and to make corrections if necessary.  For example on line 41, ref. 15 should be 14, and on line 44 ref. 16 should be 15.

2. On line 45 CIS should be defined.

3. On line 105 in the figure caption, In/InGaAs should be InP/InGaAs.

4. On line 115 RIE should be defined.

5. In Fig. 4 the caption should state which thickness of InP is present for the black curve.

6. In Fig. 5(b) it is difficult to see the difference in the red box compared to the rest of the area.  Maybe there is a way to adjust the picture to enhance the difference.

7. In Fig. 6, the question is, are curves of current density measured on one pixel of the array, or are they averages of many pixels, or are they from a separate test device with a larger area?

8. In lines 181 and 182, is this leakage on the same side of the device as the Zn diffusion?

Reviewer 2 Report

Comments and Suggestions for Authors

This paper described a method to increase the bandwidth of FPA detectors by removing part of the InP layer to reduce the reflection loss. It is a straightforward modification and demonstrates the increased quantum efficiency at the UV band.  This technique would simplify some applications that require multiple detectors to cover wide bandwidth. 

Some minor correction suggestions in the discussion section:

-Line 153-154:  “By contrast….”.  Needs native speaker English check

-3.2. It is not satisfactory that the author just gave numbers.  It would be good if the author indicated whether the increased non-uniformity was acceptable.  Same for the dark current comparison. 

-Operability: Did the authors only have one sample to work with?  What would be the repeatability of the quality of the samples? 

Comments on the Quality of English Language

English is good. Needs minor editing

Reviewer 3 Report

Comments and Suggestions for Authors

 This work reports an ultra-broadband UV-VIS-SWIR 640×512 15 μm InGaAs focal plane array (FPA) by removing the n-InP contact layer in the active area and reserving the InP contact layer around the pixels for n contact, creating incident light to be directly absorbed by the In0.53Ga0.47As absorption layer. Theoretically, the optical absorption properties of InGaAs infrared detectors with and without an n-InP contact layer are studied. The InGaAs FPA detectors have widespread application in both civil and military fields. However, more scientific details have to be firmly addressed, and I recommend the work to be accepted after major revisions by the following points.

1.        The figure resolution is a little insufficient, please improve them, such as Figure 3.

2.        The detectivity and response time are critical parameters of photodetectors, I advise the authors add them.

3.        Regarding to the imaging effect, I advise the authors use a mask with some distinguished pattern.

4.        Figure 2 seems somewhat deformed, please modify it.

5.        Figure 5a and Figure 5b seems the same image, I advise the authors revise them, despite the n-InP contact etching can be recognized after magnification.

6.        The device manufacture procedures are not discussed clearly, I advise the authors add them.

7.        Figure 8a and Figure 8b seems the same image, I advise the authors revise them carefully.

8.        The reference styles consistent with the Journal requirements, please carefully check them.

Comments on the Quality of English Language

The English Language can be further improved. 

Round 2

Reviewer 3 Report

Comments and Suggestions for Authors

The revisions are fine, it can be accepted.
